# Endocrine-Disrupting Chemicals and Early Puberty in Girls

**DOI:** 10.3390/children8060492

**Published:** 2021-06-10

**Authors:** Anastasios Papadimitriou, Dimitrios T Papadimitriou

**Affiliations:** 1Pediatric Endocrinology Unit, Third Department of Pediatrics, National and Kapodistrian University of Athens, “Attikon” University Hospital, Haidari, 12462 Athens, Greece; 2Pediatric—Adolescent Endocrinology and Diabetes, Athens Medical Center, 15125 Marousi, Greece; info@pedoendo.net; 3Endocrine Unit, Aretaeion University Hospital, 11528 Athens, Greece

**Keywords:** puberty, endocrine-disrupting chemicals, girls, precocious puberty, early puberty, constitutional advancement of growth

## Abstract

In recent decades, pubertal onset in girls has been considered to occur at an earlier age than previously. Exposure to endocrine-disrupting chemicals (EDCs) has been associated with alterations in pubertal timing, with several reports suggesting that EDCs may have a role in the secular trend in pubertal maturation, at least in girls. However, relevant studies give inconsistent results. On the other hand, the majority of girls with idiopathic precocious or early puberty present the growth pattern of constitutional advancement of growth (CAG), i.e., growth acceleration soon after birth. Herein, we show that the growth pattern of CAG is unrelated to exposure to endocrine-disrupting chemicals and is the major determinant of precocious or early puberty. Presented data suggest that EDCs, at most, have a minor effect on the timing of pubertal onset in girls.

## 1. Introduction

Pubertal timing is multifactorial involving a predominant effect of genetic and epigenetic factors and to a lesser—but still significant—extent, environmental factors. Although genetic factors are considered to explain 50–80% of the onset of puberty [1], genes that have been found to play a role in pubertal onset have a minor role in the onset of puberty of the population. Gene mutations, e.g., *ESR1*, *KISS1*, *KISSR1*, *MKRN3*, are rarely identified as the cause of disordered pubertal timing; thus, these genes do not seem to determine the timing of puberty nor menarche in the female population [2]. Recently, it was shown that the onset of puberty is regulated by epigenetic mechanisms, Kiss1 expression is negatively regulated by two polycomb group proteins (*Cbx7 and Eed*) [3].

Environmental factors are major determinants of the onset of puberty and the age at menarche. Nutrition of the mother or/and of the infant [4], chronic diseases and chronic somatic stress, like strenuous exercise or psychological stress, e.g., violence exposure [5], or adoption of a girl from an underprivileged environment, exert a major influence on the timing of the pubertal events. Although environmental factors may result in epigenetic modifications in an organism, the epigenetic effects of the environment on the hypothalamic regulation of puberty are still to be discovered.

In girls, when pubertal onset occurs before the age of 8 years, it is considered precocious and when it occurs after 8 years but before 9 years of age, it is considered early. Precocious puberty (PP) when it is gonadotropin dependent is called central or true PP and when it is gonadotropin independent is called peripheral PP. The causes of central precocious puberty may be organic, e.g., due to tumors of the central nervous system (CNS) or most commonly idiopathic, i.e., no etiological factor is identified with appropriate imaging of the CNS. Obviously, it is important to differentiate between organic or idiopathic precocious puberty (IPP) because the former may have dire health consequences. On the other hand, early puberty lies on the extreme of normal variation of timing of pubertal onset [6].

In recent decades, there have been reports from several countries detailing that the onset of puberty in girls occurs at a younger age than previously [7,8]. At the same time, endocrine-disrupting chemicals (EDC) have been suggested as affecting the age of pubertal onset, especially in girls. Hence, researchers were led to hypothesize that increasing exposure to EDC had a role in the secular trend for earlier sexual maturation. Moreover, it was suggested that early puberty manifesting in immigrants from developing countries was the result of previous exposure to organochloride pesticides [9].

Constitutional advancement of growth (CAG) is the growth pattern of early growth acceleration, which is present in the majority of girls with idiopathic precocious puberty, and in girls with early puberty [6,10]. Herein, we show that CAG is unrelated to EDC exposure and we maintain that EDCs have, at most, a minor role on female pubertal timing.

## 2. Endocrine-Disrupting Chemicals

Endocrine-disrupting chemicals are compounds that can interfere with the activity of endocrine systems. EDCs action is exerted by imitating or blocking hormone signaling through the relevant hormonal receptor. EDCs may also modulate the synthesis, metabolism, and binding of natural hormones.

EDCs are usually used by industry, as plastics (bisphenol A (BPA)), plasticizers (phthalates), solvents/lubricants (polybrominated biphenyls (PBBs), polychlorinated biphenyls (PCBs), dioxins), pesticides (chlorpyrifos, dichlorodiphenyltrichloroethane (DDT), methoxychlor), fungicides (vinclozolin) and also as flame retardant additives in manufactured materials and pharmaceutical agents, e.g., diethylstilbestrol (DES), a non-steroidal synthetic estrogen [11].

EDCs may also be made by nature, e.g., phytoestrogens, which interfere with endogenous endocrine function, are produced by plants and act primarily through estrogen receptors [12].

The abundance of EDCs and their ability to interfere with the endocrine system combined with the secular trend for earlier onset of puberty has led many researchers to associate EDCs with early puberty, especially since some EDCs have estrogenic activity.

## 3. Association between Exposure to Endocrine-Disrupting Chemicals and Timing of Puberty

Commonly used and studied EDCs are phthalates, bisphenol, pesticides and flame retardants.

**Phthalates** are mainly used as plasticizers, i.e., substances added to plastics to increase their flexibility, transparency, durability, and longevity. They are used primarily to soften polyvinyl chloride (PVC). Human exposure to phthalates [13] is extremely prevalent, and it may occur through multiple routes: i.e., oral (via phthalate-contaminated food, water and other liquids and in children through mouthing of toys and teethers), dermal (via cosmetics and other personal care products), inhaled (from breathing phthalate particles in the air) (https://www.cdc.gov/biomonitoring/Phthalates_FactSheet.html accessed 5 June 2021). Phthalates may have anti-androgenic activity [14], and also possess some estrogenic activity [15]. In Puerto Rico, high phthalate levels have been linked to premature thelarche [16], a normal variant of female premature sexual development. In a study of Danish schoolgirls, high phthalate excretion in urine was associated with delayed pubarche, but not thelarche, which suggests anti-androgenic actions of phthalate [17]. Similar results were obtained in a study of US girls [18]. Some studies reporting early/precocious puberty associated with phthalate exposure [19,20]**,** whereas in a US study on girls with central precocious puberty, such an association was not found [21]. In contrast to the previous reports, in a Chinese study performed in boys, elevated levels of phthalates were associated with constitutional delay of growth and puberty [22]. Furthermore, a recent Korean study showed that phthalate metabolites in girls with central precocious puberty were significantly lower than the prepubertal control girls [23]. Therefore, more studies are warranted to explore the effect of phthalate exposure on pubertal timing.

**BPA** (bisphenol A) is found in plastics (e.g., bottles, Tupperware, etc.), and in epoxy resins coating the inside of beverage and food cans, and humans are exposed mainly through food contamination from plastic and can packaging. BPA is the most commonly found estrogen-like endocrine disruptor in the environment. In experimental animals, it has been shown that BPA advances puberty [24], but on the other hand, it also has been shown it has no effect on pubertal timing [25]. Similar to the experimental animals, results of BPA on human puberty are inconsistent. In a US study of girls, Wolff et al. reported that BPA had no influence on breast development [26], however in studies performed in Turkey and in Thailand, idiopathic central precocious puberty was associated with higher levels of BPA than in control girls [27,28]. Watkins et al. studied the in-utero and peripubertal exposure to phthalates and BPA in relation to sexual maturation and did not find any association between BPA and sexual maturation, although in utero phthalate exposure impacted on earlier timing of sexual maturation [29].

**Pesticides** are classified into various classes, e.g., insecticides, herbicides and fungicides. Pesticides enter the human body through water, air, food, and also, they can pass from mother to fetus via the placenta and to the infant through mother’s milk. Most of these substances are lipophilic and accumulate in adipose tissue, where they remain for long periods of time. The most widely studied pesticide compounds are dichlorodiphenyltrichloroethane (DDT) and its metabolite dichlorodiethyldichloroethane (DDE). Although the agricultural use of DDT has been banned worldwide, in some developing countries, it is still used against mosquitos. In a study performed in Denmark, mothers who worked in greenhouses in the first trimester of pregnancy were prenatally categorized as exposed or unexposed to pesticides. Female offspring of exposed mothers had decreased age of breast development at 8.9 years, compared with 10.4 years in the unexposed, and 10.0 years in a Danish reference population [30]. Other researchers reported decreased menarcheal age to girls exposed in utero to DDE [31]. However, the significance of the association disappeared when weight at menarche was controlled for. Adopted or immigrant girls in Belgium, who presented central precocious puberty (CPP), had increased plasma levels of pesticides (DDE), thus CPP could be attributed to pesticide exposure [9]. However, other studies did not find an association between DDE levels and the timing of menarche [32]. Additionally, in inner-city girls, DDE, Pb, and dietary intakes of phytoestrogens were not significantly associated with breast stage [33]. In a study examining prenatal DDT exposure in relation to anthropometric and pubertal measures in adolescent males, no associations between prenatal exposure to any of the DDT compounds and any pubertal measures were noted [34].

**Flame-retardant chemicals are** added to manufactured materials (plastics, textiles, surface finishes and coatings) intended to prevent or slow the further development of ignition with their physical and chemical properties. Among them, organohalogen compounds such as polybrominated diphenyl ethers (PBDEs) are lipophilic persistent endocrine disruptors exhibiting estrogenic as well as androgenic properties. It has been proposed that PBDEs might alter pubertal timing resulting in later menarche in girls [35,36] but earlier pubarche in boys [36]. Curiously, girls with idiopathic central precocious puberty, particularly those with higher body mass index (BMI) have been found with higher serum concentrations of PBDEs [37].

Thus, the inconsistency of the results of the various studies examining the association of endocrine disruptor chemicals with the onset of puberty [38] makes it imperative that more studies on the subject are performed.

## 4. Constitutional Advancement of Growth

Constitutional advancement of growth (CAG) is a growth pattern characterized by early growth acceleration [10]. Children with CAG are born with an average length but present growth acceleration soon after birth, reaching a zenith centile in the first 2 to 4 years of life. Then, the child grows along this centile until the onset of puberty, which is usually early. The growth pattern of CAG is the mirror image of the well-established growth pattern of constitutional delay of growth (CDG), which is characterized by growth deceleration in the first 2 to 3 years of life, although birth length is usually average. As a result of the growth deceleration, the child’s height may fall to a nadir centile, which is at or below the third centile, depending on parental height. Then, growth resumes at a normal rate, and the child grows along this centile until the onset of puberty, which is usually delayed.

For a child to be considered as presenting the growth pattern of CAG, his/her Height Standard Deviation Score (SDS) should be ≥1.5 than Target Height (TH) SDS and other conditions that lead to early growth acceleration, such as genetic tall stature, overfeeding, and intrauterine growth restraint have to be excluded. In the case of a typical girl, the growth pattern of CAG is depicted in Figure 1.

Red bullets depict actual height, orange squares depict bone age. Target Height (TH) derived from the heights of the mother (yellow dot) and father (blue dot) is shown on the right-hand side of the figure and as a blue line within the graph.

## 5. Constitutional Advancement of Growth and Early Puberty

Based on the observation that girls with precocious puberty (i.e., with breast development before 8 years of age) are tall for age, and also taller for TH, even at the onset of pubertal development, we examined the growth of 47 girls with IPP from birth until diagnosis [39]. The vast majority, i.e., 79%, of the girls with IPP manifested the growth pattern of CAG. Thus, we suggested that this pattern may be used as an additional clue in favor of idiopathic precocious puberty in the differential diagnosis of precocious puberty [10], a claim that was supported by the fact that all girls with CAG presented no CNS abnormalities in brain magnetic resonance imaging (MRI) [39]. We also observed a similar growth pattern in girls with early puberty, i.e., with breast development at an age between 8 and 9 years [40]. The greater the difference HSDS-THSDS is above +1.5 SD the greater the possibility for the girl to present precocious than early puberty.

## 6. Is There an Association between EDC Exposure and CAG?

Since CAG is associated with early growth acceleration, if there was an association between EDCs and CAG it would be related to fetal exposure during pregnancy or early postnatal exposure.

Several studies have examined the association of fetal exposure to EDCs, especially phthalates and BPA, and fetal growth. These compounds, besides hormonal perturbations, may cause oxidative stress and epigenetic modifications that might have a deleterious effect on fetal growth [41,42,43]. Most relevant studies were cross-sectional, measurements of length and weight were performed at delivery [44] and the results were inconsistent [45,46,47]. In a longitudinal study that examined the relationship between average exposure measures and fetal growth [48], researchers observed inverse associations between head and abdominal circumferences, femur length, and estimated fetal weight and Di (2-ethylhexyl) phthalate (DEHP) metabolites. However, no consistent associations were observed for other phthalate metabolites or for BPA. Taken together, these data indicate that it is unlikely EDC exposure in utero has a substantial adverse effect on fetal growth. In our studies on CAG girls, there was no difference in birth weight or length compared to control girls.

What about the effect of postnatal exposure on growth? In a study in which chlordecone (an organochlorine insecticide with estrogenic properties) was measured at cord blood and in breast milk at the age of 3 months, postnatal exposure in girls was associated with lower height at 3, 8 and 18 months [49].

Taking into account that girls with CAG present growth acceleration soon after birth, it is unlikely that growth acceleration is induced by estradiol. In neonatal life and early infancy increased estrogen levels are experienced by all normal girls. Estradiol levels peak around the second month after birth and reach levels characteristic of early- to mid-puberty, hence the period of the first 6 months of life is termed as mini-puberty. Despite increased estradiol levels, mini-puberty is not associated with growth acceleration contrary to puberty occurring during childhood.

How could the early growth acceleration characteristic of CAG be explained? Clinical evidence suggests that after the age of 2 to 3 months growth hormone (GH) is necessary for normal growth [50]. However, the majority of small for gestational age (SGA) infants present catch-up growth in length since early postnatal life placing them at or above the 3d percentile for length by 6 months of age [51]. Catch-up growth in these children is mostly attributed to overnutrition, thereby increasing insulin levels, a hormone with growth-promoting properties. However, from the 3d day of life SGA neonates present functional hypersomatotropism, i.e., increased GH and insulin growth factor 1 (IGF-1) levels relative to appropriate for gestational age neonates, suggesting that the somatotropic axis is fully operational since the first days of life [52,53]. In line with a functional GH/IGF-1 axis from the first days of life is the observation that, in healthy full-term neonates, the postnatal growth velocity is positively related to a spurt in immediate postnatal life IGF-1 levels [54]. It is noteworthy that the increased IGF-1 levels may persist in later years [55].

SGA children presenting catch-up growth are more prone to insulin resistance and development of metabolic syndrome [56]. Recently a study on Sprague–Dawley rats examined whether a post-receptor crosstalk of GH and insulin signaling might affect insulin resistance in catch-up growth SGA animals [57]. The authors demonstrated that catch-up growth SGA rats exhibit increased insulin resistance associated with an impaired IRS-1-PI3K-AKT signaling pathway, which resulted from GH signaling-induced upregulation of SOCS3 expression. Thus, these data suggest a link between increased GH levels and insulin resistance in catch-up growth.

Constitutional advancement of growth presents similarities to catch-up growth only that the CAG children are, auxologically, appropriate for gestational age. Accordingly, they are susceptible to developing obesity during childhood, therefore we suggested that the growth pattern of CAG may be a predictor, not only of early puberty, but of childhood obesity as well [10]. Moreover, it has been reported that earlier menarche was associated with greater height, adiposity, and significantly increased serum IGF-1 at 8 years of age, even after adjustment for height and BMI [58]. Thus, these data allow us to speculate that the growth pattern of CAG is induced by the early activation of the GH/IGF-1 axis.

## 7. Concluding Remarks

The secular trend of pubertal onset in girls as reflected by the age at menarche has started since the early 20th century with the improvement of the socioeconomic and hygienic conditions. This trend, however, has levelled off in many countries of the developed world [59] and it seems that it has reached the age range of menarche occurrence in the paleolithic female [60], suggesting that human females evolved to enter puberty at a young age.

There is a growing body of evidence that some EDCs alter the metabolic set-points and increase the risk of obesity [61], and that EDC exposure during fetal life is associated with breast cancer [62]. It is of interest that the increase in obesity prevalence coincides with increases in endocrine-disrupting chemical production and exposures [63]. The rapid increase in the prevalence of obesity in the last three decades suggests that environmental factors are major players in the obesity epidemic. The association between EDCs and early onset of puberty not only is inconsistent but is complicated by the fact that, at least in girls, pubertal events may be advanced by obesity [64].

Undoubtedly, there is a great need for the EDC effects on the human body systems to be studied thoroughly. However, from the data presented in this review, it is clear that the major determinant of early puberty, at least in girls, is the presentation of the growth pattern of constitutional advancement of growth, which is unrelated to EDC exposure. Therefore, if there is a role of EDCs on female pubertal timing it seems, at the most, to be a minor one.

## Figures and Tables

**Figure 1 children-08-00492-f001:**
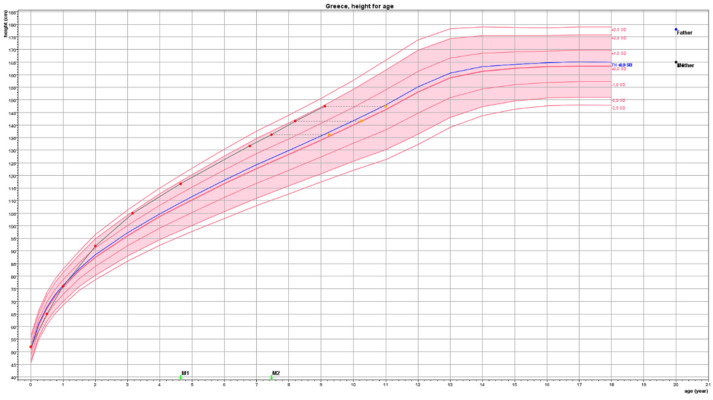
This girl presented at the clinic at the age of 8.1 years with a Tanner stage 2 breast and pubic hair development. Breast developed at the age of 7.4 years and pubic hair at the age of 7.8 years. Before the appearance of any signs of puberty, her height SDS (HSDS) was +1.84 SDS (lines in growth chart represent height standard deviations marked as SD) well above her Target Height (TH) of +0.25 SDS. She was born with an average birth length of 0.0 SDS, but soon after birth, she presented growth acceleration reaching a zenith percentile (close to HSDS + 2 SDS) at about 3 years of age. She continued to grow along this percentile until she entered puberty presenting a further growth acceleration thereafter. At the age of 8.2 years, her bone age was 10.3 years compatible with the biological age in which girls enter puberty. Predicted adult height was within TH. No treatment was considered necessary.

## Data Availability

Not applicable.

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
