# Peer review of "Endocrine-Disrupting Chemicals and Early Puberty in Girls"

_children, 2021, doi:10.3390/children8060492_

Round 1
Reviewer 1 Report
In recent decades the onset of puberty in girls was often reported at a younger age than previously. It was hypothesized that increasing exposure to endocrine disrupting chemicals (EDC) had a role in the secular trend for earlier sexual maturation. Moreover, constitutional advancement of growth (CAG) is present in the majority of girls with idiopathic precocious and early puberty. Aim of this Review was to analyze the current views, based on literature data, about the relationship between EDC, CAG and precocious puberty in girls. In the present Review the Authors showed that CAG could be unrelated to EDC exposure and EDCs might have a minor role on female pubertal timing.
I have read this Paper with great interest. It is well written. The topic is very interesting, very important and very “hot”, especially from a clinical point of view. The literature search is adequate.
It is not a systematic review, as per the editorial rules for the "Children" journal. It is a narrative review. But a more adequate definition for this Paper could be as a Commentary.
Reviewer 2 Report
This review aimed to explore the potential role of endocrine disrupting chemicals in triggering precocious and early puberty in girls. The authors reviewed studies focusing on the association between exposure to phthalates, bisphenol A and pesticides and timing of puberty but did not consider those on exposure to flame retardants. Some examples of these studies are listed below:
- Harley KG, Rauch SA, Chevrier J, Kogut K, Parra KL, Trujillo C, Lustig RH, Greenspan LC, Sjödin A, Bradman A, Eskenazi B. Association of prenatal and childhood PBDE exposure with timing of puberty in boys and girls. Environ Int. 2017 Mar;100:132-138. doi: 10.1016/j.envint.2017.01.003
- Tassinari R, Mancini FR, Mantovani A, Busani L, Maranghi F. Pilot study on the dietary habits and lifestyles of girls with idiopathic precocious puberty from the city of Rome: potential impact of exposure to flame retardant polybrominated diphenyl ethers. J Pediatr Endocrinol Metab. 2015 Nov 1;28(11-12):1369-72. doi: 10.1515/jpem-2015-0116
- Windham GC, Pinney SM, Voss RW, Sjödin A, Biro FM, Greenspan LC, Stewart S, Hiatt RA, Kushi LH. Brominated Flame Retardants and Other Persistent Organohalogenated Compounds in Relation to Timing of Puberty in a Longitudinal Study of Girls. Environ Health Perspect. 2015 Oct;123(10):1046-52. doi: 10.1289/ehp.1408778
Abbreviations should be mentioned at the first appearance along with the full name. Consequently, the list of abbreviations is currently incomplete.
Line 55. Consider replacing “that” with “which”.
Line 69. Please see the previous comment.
Line 79. Please provide a brief description of the uses of phthalates and their routes of exposure to humans.
Line 105. Please add “in” before “relation”.
Line 148. Please provide the full name of “SDS”.
Line 149. Consider replacing “like” with “such as”.
Line 158. Please provide the full name of “HSDS”.
Line 157. Please give here the abbreviation of Target Height.
Figure 1 is unclear. What does each line represent? Also, it would take more contrast between the colors.
Line 269. Please provide the full name of IPP.
Line 185. The authors wrote “most relevant studies were cross-sectional”, but they cited only one study of them. Please provide more references.
Line 199. This statement seems to be in contrast with what was declared in lines 192-194.
Line 207. Please provide the full name of “GH”.
Line 213. Please provide the full name of “IGF-1”.
Line 237. “at age” was repeated twice.
Line 246. This sentence needs appropriate references.
Round 2
Reviewer 2 Report
The authors addressed all my previous concerns and the paper is now acceptable for publication.
Minor comments.
Line 84. Please add a parenthesis after “oral”.
Line 85. Consider deleting “or” and adding a comma after “teethers”.
Line 146. “ethers”, not “ether”. Accordingly, please use the acronym “PBDEs” instead of “PBDE”.
Lines 149-150. Consider deleting “though” and adding “those” after “particularly”.
Author Response
The authors are grateful to reviewer 2 for helping them to improve their manuscript substantially. All her/his suggestions were addressed.